

# Developing a low-cost frequency-domain electromagnetic induction instrument

Gavin Wilson[1], Jacob Conrad[1], John Anderson[1], Andrei Swidinsky[1,2], and Jeffrey Shragge[1]

[1]Department of Geophysics, Colorado School of Mines, Golden, CO, United States
[2]Now at Department of Earth Sciences, University of Toronto, Toronto, ON, Canada

**Correspondence:** Gavin Wilson (gswilson@mines.edu)

**Abstract.** Recent advancements and the widespread availability of low-cost microcontrollers and electronic components have created new opportunities for developing and using low-cost, open-source instrumentation for near-surface geophysical investigations. Geophysical methods that do not require ground contact, such as frequency-domain electromagnetics, allow one or two users to quickly acquire large amounts of ground resistivity data. The Colorado School of Mines electromagnetic system (CSM-EM) is a proof-of-concept instrument capable of sensing conductive objects in near-surface environments, is similar in concept to commercial grade equipment, and costs under US$400 to build. We tested the functionality of the CSM-EM system in a controlled laboratory setting during the design phase and validated it over a conductive target in an outdoor environment. The transmitter antenna can generate a current of over 2.5 A, generating signals that are detectable by a receiver antenna at offsets of up to 25 m. The system requires little refitting to change the functioning frequency, and has been operationally validated at 0.4 kHz and 1.6 kHz. The receiver signal can be measured by off-the-shelf digital multimeters. Future directions will focus on improving the electronic and mechanical stability of the CSM-EM with the goal of using acquired data to make quantitative estimates of subsurface resistivity distribution.

## 1  Introduction

Near-surface geophysical surveying using electrical and electromagnetic (EM) methods has experienced growth in recent years due to increased interest in identifying groundwater resources (Boaga, 2017), addressing environmental remediation (White et al., 2016), and performing archaeological reconnaissance studies (Sea and Ernenwein, 2020). Near-surface geophysical techniques provide non-invasive and cost-effective approaches for imaging subsurface structures and estimating earth properties, compared to methods such as drilling or excavation (Ward and Hohmann, 1988). Additional near-surface EM application examples include using conductivity data combined with soil sampling and satellite imagery to develop frameworks for farm irrigation management (Fontaine et al., 2018); applying EM methods for environmental remediation to assess the location and areal extent of pollutants including landfills and radioactive waste disposal sites (White et al., 2016); and developing light-weight drone-based EM systems for detecting and classifying unexploded ordinance (Shubitidze et al., 2021), which opens up opportunities for drone-based EM surveying in environmental and agricultural applications.





While geophysical methods can assist with subsurface investigations, the cost of commercial instruments required to perform such surveys can be prohibitively expensive and form an effective "barrier to entry" for many potential users. The price constraints of many commercial grade instruments stem from their hardware being designed for large-scale campaigns, industrial applications, and the capability to acquire high-quality data under extreme climate conditions (e.g., from the frozen Arctic to the hot desert). This leads to scenarios where commercial instruments are effectively "over-engineered" for many near-surface geophysical applications, when more elementary instrumentation and data acquisition procedures would suffice.

The recent rapid growth of low-cost microcontrollers (e.g., Arduino and Raspberry Pi) and sensors as well as the proliferation of open-source software packages allow entry-level and expert practitioners alike to build high-accuracy sensor systems at a price range affordable for small-scale research and enthusiast projects. These tools have the potential to be leveraged in purpose-built low-cost geophysical equipment that can acquire data without exceeding the durability and budgetary constraints for many types of near-surface geophysical investigations. Examples following this low-cost instrumentation approach include direct-current (DC) resistivity (Clark et al.; Ahmad et al., 2019; Sirota et al., 2021), seismic nodes (Dean et al., 2017; Soler-Llorens et al., 2019; Wilson et al., 2021), and magnetometers (Schofield et al., 2012; Shahsavani, 2018), each of which has demonstrated the possibility of acquiring data of comparable quality to commercial grade systems. While such home-grown instrumentation is neither as robust nor as likely to have the in-built safety factors as commercial grade instruments, it can lower the barrier-to-entry for many users, enable enthusiast or humanitarian geoscience applications, and be used to develop low-cost geophysical networks for time-lapse monitoring projects.

Frequency-domain electromagnetic methods (FDEM) represent a class of geophysical techniques that are important for near-surface applications due to their sensitivity to subsurface variations in electrical resistivity (or its inverse, electrical conductivity) due to, e.g., heterogeneity in geological material or variable fluid saturations. FDEM surveying is based on the principle of electromagnetic induction and requires only one or two operators to acquire data, meaning that the instrument does not have to be attached to the earth unlike grounded methods (e.g., DC resistivity, induced polarization, and seismic). This advantage allows users to acquire spatial FDEM geophysical data at greater rates than comparable ground-coupled methods, and furthermore makes the approach a strong candidate for EM drone-receiver based investigations. Overall, developing a low-cost FDEM system prototype that provides accurate data could create significant opportunities for numerous near-surface geophysical applications.

The primary goal of this study is to design, build and validate a low-cost transmitter-receiver FDEM system for under US\$500. Our developed proof-of-concept instrument, the CSM-EM, is of comparable size and transmits similar signal strength to the commercial Geonics EM-34 system, and can detect conductive objects using amplitude-based signal measurements via an autoranging digital multimeter (DMM). The instrument is straightforward to operate and can function at a variety of transmitter/receiver frequencies with minimal refitting. The paper starts by briefly describing the theory behind the FDEM method and its use in geophysical investigations. We then discuss our low-cost FDEM transmitter-receiver system design and provide details on its construction. We conclude by presenting validation results for the system prototype in both laboratory and outdoor conditions, and by discussing the cost breakdown and future refinements of the device, along with possible applications in near-surface geophysical investigations.



## 2 Methods

The purpose of this section is to provide a brief review of the theoretical principles and some methodological considerations behind the FDEM technique. Readers interested in a more complete theoretical treatment are referred to standard reference texts on these subjects (Ward and Hohmann, 1988), (Purcell, 1966).

### 2.1 FDEM Theory Overview

Electromagnetic methods measure ground resistivity through EM induction with separated transmitter (Tx) and receiver (Rx)
antennas, allowing data acquisition without the need for ground contact. As detailed in the Instrument Design section below, the two antennas are circular coils of wire connected to different electronic modules. FDEM methods are based upon the principles of Ampere's and Faraday's Laws in a quasi-static regime, where an alternating current (AC) produces a magnetic field and an alternating magnetic field produces an electric field, respectively. As illustrated in Figure 1, FDEM uses a known and calibrated time-varying current in the Tx coil (blue loop) to produce an alternating primary magnetic field that is present
in both the air and subsurface (solid gray lines). This field operates at a single frequency specified by the user. For scenarios involving conductive subsurface material, this alternating magnetic field will induce alternating eddy currents (orange lines) via Faraday's Law. As described by Ampere's Law, these eddy currents produce a secondary magnetic field (dashed black lines). The superimposed fields produce a time-varying current in the Rx coil (red loop), as per Faraday's Law. This signal will oscillate at the same frequency as the field generated by the Tx and can be measured as a voltage drop across a capacitor
attached in series with the Rx antenna coil.

When the system is over a uniform halfspace of a given conductivity, a Tx-Rx configuration with a fixed offset and orientation will measure a constant voltage independent of lateral position. However, the presence of lateral subsurface heterogeneities will affect the signal detected by the Rx and introduce spatial dependence in the voltage measurements. This makes FDEM particularly effective at identifying lateral changes in soil-water content and at locating anomalous conductive bodies (such as
illustrated in Figure 1). Some commercial EM systems also use a tuneable third "bucking" coil to remove the primary field response from the Rx coil, due to the large amplitude of the primary field compared to that of the secondary field (the latter of which contains the relevant information about ground conductivity).

### 2.2 Penetration Depth

The penetration depth of any EM method is dependent upon a variety of subsurface physical properties as well as the frequency
of the inducing magnetic field produced by the Tx. For FDEM field applications, the effect of these properties can be quantified using a proxy value known as the skin depth $\delta$

$$\delta = \frac{1}{\sqrt{\pi \mu \sigma f}}, \tag{1}$$

which represents the depth in the subsurface at which the field strength has decayed to $1/e$ (37%) of the surface value. Skin depth is inversely proportional to the square root of the subsurface electrical conductivity $\sigma$, magnetic permeability $\mu$, and the





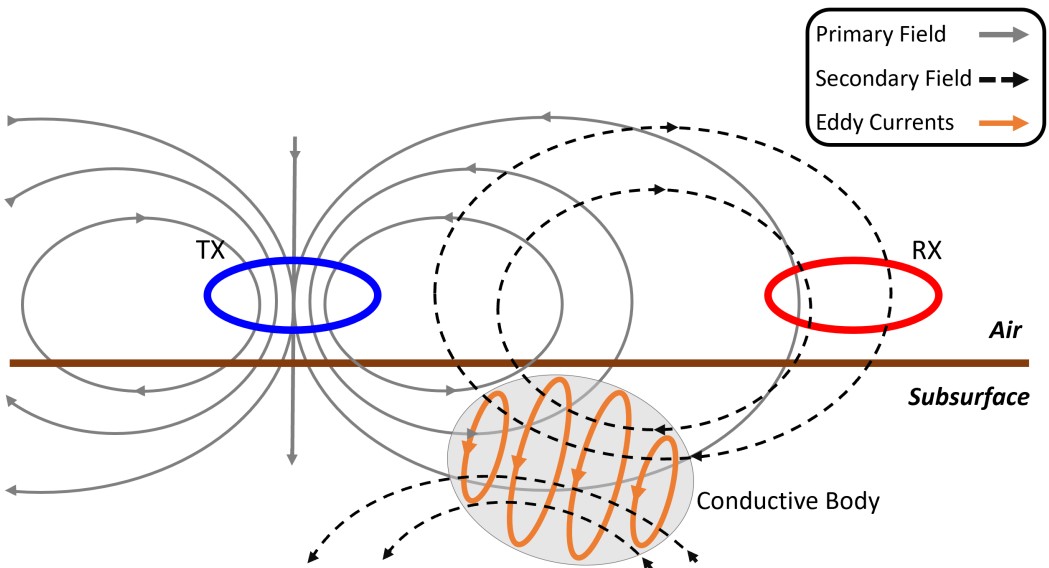

**Figure 1.** The magnetic fields resulting from a horizontal coplanar Tx-Rx system configuration over a conductive body. The Tx electronics produce a time-varying current in the Tx coil (blue ring) that generates an alternating magnetic field (solid light gray line) that is present in the air and subsurface. This field will generate eddy currents in conductive subsurface material (orange lines) that create a secondary magnetic field (black dashed lines). The Rx unit measures the combined effects of the time-varying primary and secondary magnetic fields as a voltage.

operating frequency of the instrument, $f$. Electrical conductivity is related directly to subsurface geological structure, as well as any fluids that might be saturating the associated pore space. In most geological situations, the magnetic permeability $\mu$ can be assumed to equal the magnetic permeability of free space $\mu_0 = 4\pi \cdot 10^{-7}$ H/m. This leaves frequency as the only tuneable experimental variable in a given survey, meaning that the depth of investigation can be increased or decreased by respectively decreasing or increasing the Tx frequency, along with Tx-Rx offset.

While the frequency of the EM system can be used to alter the penetration depth of the inducing fields, different relative orientations of the Tx and Rx coils can be used to change the EM field radiation patterns and achieve different subsurface sensitivities. Figure 2 illustrates three common Tx-Rx "fully coupled" orientations, with the horizontal coplanar (HCP) being the most common investigation setup and the one used in the validation tests reported herein.

The Tx-Rx offset $r$ has a major effect on signal decay. For a uniform halfspace of conductivity $\sigma$ and an HCP configuration,

the signal of frequency $f$ is related to the vertical magnetic field strength detected by the Rx coil (Ward and Hohmann, 1988, Eq. 4.56). Changes in the vertical magnetic field can be measured as a voltage by the Rx, $V_R$, and is related to offset $r$ by

$$V_R = \frac{if\mu_0 A_T A_R N_T N_R I_T}{k^2 r^5}\left[9 - (9 + 9kr + 4k^2r^2 + k^3r^3)e^{-kr}\right], \tag{2}$$





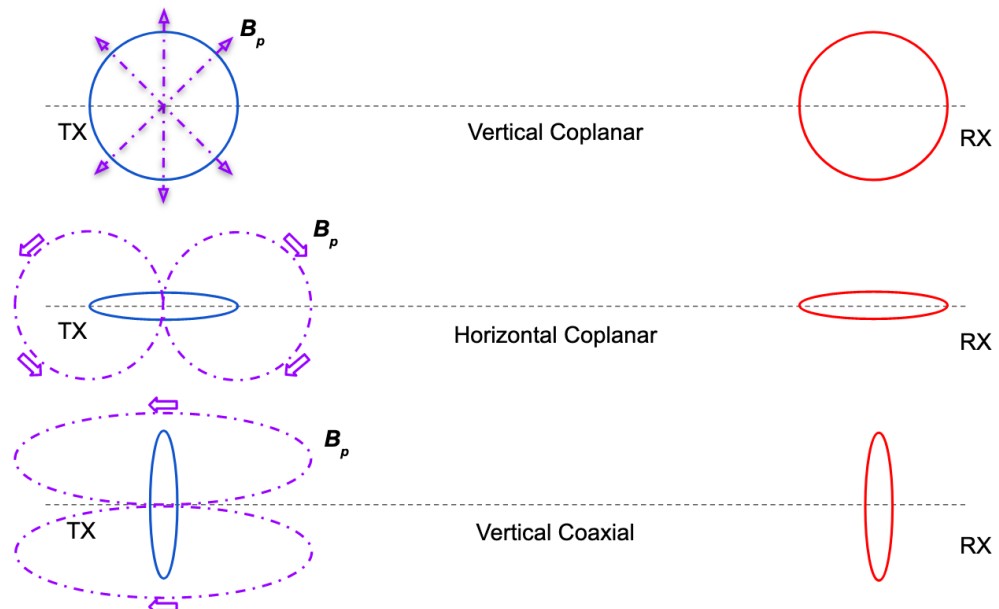

**Figure 2.** Fully coupled orientations for coplanar/coaxial Tx and Rx loops. The purple lines and arrows represent the magnetic field generated by the Tx antenna. Top: vertical coplanar. Middle: horizontal coplanar. Bottom: vertical coaxial.

where $k^2 = -i\sigma\mu_0 2\pi f$, $I_T$ is the Tx current, and $A_T$, $A_R$, $N_T$, and $N_R$ respectively are the areas and the number of wire turns around the Tx and Rx antennas. While $f$, $k$, and $r$ solely account for the magnetic field strength at the Rx coil in an HCP Tx-Rx orientation, $I_T$, $A_T$, $A_R$, $N_T$, and $N_R$ (Tx/Rx component parameters) affect both the magnetic field strength, and the value measured across Rx by a DMM.

## 2.3 Instrument Sensitivity

The choice of component parameters (i.e., the effective areas of the Tx and Rx antennas) were based off of the dimensions of the EM-34 conductivity meter due to its in-house availability and large size. Before building the prototype, the antenna parameters were tested by modeling the Rx voltage over a range of half-space resistivity values to determine whether the CSM-EM range and resolution (0.1 mV-9.0 V) allowed the instrument to delineate between different geologic environments. This testing modeled the change in Rx signal, which measures the strength of the total vertical magnetic field, over a half space as a function of Tx-Rx offset (Figure 3a). We also examined how the secondary field, containing the information related to ground resistivity, changed with variable resistivity for a range of Tx-Rx offsets (Figure 3b). The total field voltage response modeling (i.e., the value measured by the instrument) shows that signals over different half-space resistivity values fall within CSM-EM resolution for most offsets from 1 m to 50 m. The only half-space resistivity falling outside the CSM-EM resolution was the simulated perfect conductor ($\sigma = 1,000,000$ S/m). This exercise demonstrated that the effect of the secondary field is difficult to detect for most half-space resistivities. The secondary field response showed whether different half-space resistivities could





be delineated given the resolution of the CSM-EM system. The secondary field was calculated by subtracting the primary
(free space) field from the total field. This calculation showed that subsurface resistivity variations ranging from $10^0$ Ω·m to
$10^2$ Ω·m would be detectable within the CSM-EM resolution at most Tx-Rx offsets. However, more resistive environments
(e.g., $10^3$ Ω·m) would generate an insufficient secondary field to create a voltage change detectable by the CSM-EM, given
that most DMMs measure voltages to within a ±0.1 mV precision.

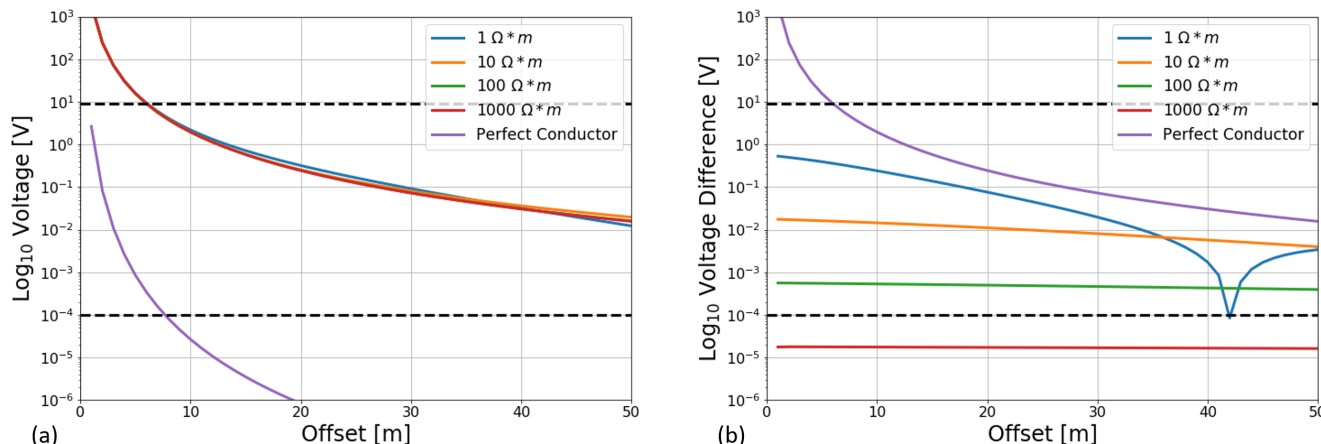

**Figure 3.** (a) The absolute value of the Rx voltage measured by a horizontal coplanar (HCP) configuration resting on a halfspace of variable
conductivity as a function of Tx-Rx offset. All Rx electronic parameters (resistance, inductance, capacitance, amplifier gain) are as indicated
in Figure 4, with a Tx frequency of 1680 Hz, a current of 2.0 A, and Tx and Rx coil areas of 0.29 m$^2$ each with 100 turns. The dashed
horizontal lines indicate the upper (9.0 V) and lower (0.1 mV) limits of the Rx amplifier power supply and DMM resolution, respectively.
(b) Difference between free-air and half-space voltages (effectively the amplitude of the secondary field) for an HCP configuration with all
parameters identical to those shown in (a).

## 3   Instrument Design

The primary goals in the design of the low-cost CSM-EM FDEM system are to: (1) construct an adaptable device able to
function at multiple signal frequencies and Tx-Rx offsets similar to commercial FDEM instrumentation; (2) use basic and
easily sourced electrical components; and (3) maintain a sub-$500 instrument build cost. While the use of low-cost components
sacrifices some of the durability found in commercial systems, easily procurable components facilitate replacement or even the
construction of multiple EM systems to work independently or jointly and at a low capital expenditure.

130       The design goals for the low-cost CSM-EM device require that the Tx-Rx frequency can adjust with minor hardware changes
and that the CSM-EM Tx signal is comparable in strength to that offered by a commercial FDEM system Tx. The Geonics
EM-34 conductivity meter system was chosen as the basis for the CSM-EM design due to in-house availability and its depth-
of-investigation relevant to near-surface applications.





### 3.1 Functionality & Workflow

The CSM-EM instrument is composed of independent Tx and Rx units. During data acquisition, the Tx hardware module generates an AC signal that is amplified and transmitted through the Tx antenna. Both the primary EM field from the Tx unit and secondary fields generated by conductive subsurface heterogeneities are measured by the Rx antenna. The composite signal is filtered and then amplified so that voltage changes can be sensed with a 0.1 mV resolution DMM (Figure 3 shows the system resolution limits) capable of measuring AC signals. The multimeter records an approximate root-mean-squared

(RMS) voltage, which is a simple measure of the signal magnitude generated by the primary Tx field and any secondary EM fields. The design and functionality of the Tx and Rx units are described below. The specific component values represented in CSM-EM design can be easily modified and are based on the calibrated inductance and resistances of the Tx and Rx coils. The resistances of the coils were measured using a DMM. Coil inductances were measured by using a known capacitor in series with the coil to create an LC circuit, scanning through a range of input frequencies until achieving resonance in the coil, and

finally back-solving for coil inductance ($L_{coil}$) using equation 5. By tailoring component values to the Tx and Rx properties, the CSM-EM can be constructed using easily substituted parts.

### 3.2 Transmitter Design

The series of hardware modules of the CSM-EM Tx unit and the circuitry design behind each module are presented in Figures 4a and 4b, respectively. The RC oscillator and power amplifier modules are powered by two 12 V motorcycle batteries.

Alternatively, the system could be powered by four 12 V batteries with two pairs connected in series to obtain a larger Tx current; however, following this approach may shorten instrument longevity in the field.

The Tx system generates an AC signal using a resistor-capacitor (RC) oscillator module. An RC oscillator generates an AC signal using EM noise, RC circuit feedback, and signal amplification (Alexander and Sadiku, 2007). Any EM noise encountered by the module is filtered through several RC stages that introduce a phase shift at a given angle depending on the chosen RC

component values and the number of stages $N_S$ within the circuit. The cumulative RC signal conditioning creates a 180° phase shift. The resulting signal is subsequently amplified by an inverting operational amplifier, which causes another 180° phase shift. This shift creates regenerative feedback, allowing for a stronger AC signal to be generated solely from ambient EM fields and a DC power source. The desired frequency $f_{RC}$ (in Hz) of the AC signal can be calculated using

$$f_{RC} = \frac{1}{2\pi RC\sqrt{2N_S}},$$ (3)

where $R$ is resistance, $C$ is capacitance, and $N_S$ is the number of RC stages. The $R$ and $C$ components in each stage of the $RC$ oscillator must have the same respective resistance and capacitance values, meaning that in Figure 4b $R = R_1 = R_2 = R_3$ and $C = C_1 = C_2 = C_3$. $RC$ oscillators commonly have three $RC$ stages to maintain signal stability (Alexander and Sadiku, 2007), which is reflected in our CSM-EM oscillator module. Representative values of $C = 1.8$ nF, R = 22.0 kΩ, with $N_S = 3$ generate an $f_{RC} = 1.64$ kHz. The output frequency of the CSM-EM system can be modified by switching out modular capacitor units

with differing $C$ values and changing the tuning capacitor attached to the Tx antenna.







**Figure 4.** (a) Low-cost CSM-EM Tx system powered by two 12 V motorcycle batteries. The RC oscillator generates an AC signal at a given frequency, which is amplified by the power amplifier before being passed into the Tx coil to generate an AC signal and corresponding magnetic field. The DMM measures the alternating current passing through the Tx coil. (b) CSM-EM Tx circuit diagram that can be split into three primary modules: the RC oscillator, the power/signal amplifier, and the Tx coil (with tuning capacitor). The circuit is composed of basic electronic components along with an OPA549 power amplifier breakout board. Resistor $R_5$ was included to allow regenerative feedback in the circuit and is not part of a specific Tx module.



The stability of the signal output also depends on the applied amplifier gain. Gain is a unitless value that describes the ratio between the voltage of the output signal from an operational amplifier (op-amp) to that of the input signal. The signal gain $G$ of the output voltage $V_{out}$ is dependent on the values of resistors $R_f$ and $R$ attached to the op-amp and on the voltage of the input signal $V_{in}$, as follows for a simple inverting amplifier:

$$V_{out} = G V_{in} = -V_{in} \left( \frac{R_f}{R} \right). \tag{4}$$

Figure 4b uses this setup within the RC oscillator assembly with a gain $G = 31$, which is controlled by resistors $R = R_3$ and $R_f = R_4$ in Figure 4b. The op-amp gain of an RC oscillator must be $G \geq 29$ to main signal stability (Alexander and Sadiku, 2007); however, a value $G \gg 29$ tends to distort the AC signal.

The small AC signal generated by the RC oscillator is impractical at any field scale Tx-Rx offset (Alexander and Sadiku, 2007). The high-voltage, high-current op-amp device (OPA549) can supply a current up to 8.0 A to any load attached to the output. The OPA549 has been driven to saturation as a $\pm 12$V square wave, the maximum signal gain allowable by the power supplies, and does not follow equation 4. This specialized power amplifier is required for signal amplification needed to generate measurable field-scale signals. The power amplifier is part of a pre-fabricated breakout board with DC power supply regulating capacitors and a heat sink. We initially built a heat sink attachment in-house (Central unit in Figure 4b), but the pre-fabricated unit (Power amplifier in Figure 4a) proved to be safer, more durable, and reliable under field conditions.

Finally, the amplified signal is passed into the Tx antenna, consisting of 100 turns of wire around a 0.61 m (2 ft) diameter coil. To generate resonance in the Tx coil at the same frequency as the RC oscillator, we must include a tuning capacitor (Alexander and Sadiku, 2007). The resulting RLC circuit uses a capacitor $C_{coil}$ and the inherent inductance $L_{coil}$ of the Tx antenna to provide an effective bandpass filter. The filtered output frequency $f_{coil}$ is inversely proportional to the square root of the $L_{coil}C_{coil}$ product:

$$f_{coil} = \frac{1}{2\pi \sqrt{L_{coil}C_{coil}}}. \tag{5}$$

The Tx antenna had a measured inductance of $L_{coil} = 13.28$ mH that when combined with a $C_{coil} = 770$ nF tuning capacitor created a bandpass filter with a peak frequency of $f_{coil} = 1.57$ kHz. The LC circuit ensures that when the square-wave signal passes through the LC circuit, it is transmitted as a monochromatic sinusoidal wave. Although LC filters theoretically pass a single frequency, the Tx wire itself has a fixed resistance value; when combined with the LC filter inductance $L_{coil}$, the resistance $R_{coil}$ dictates the filter pass-band width $\Delta f_{width}$ (i.e., the frequency range about the peak value that is still passed through the filter). Whereas too broad of a pass band would weaken the resonance of the Tx antenna, too narrow of a pass band may decrease the signal strength at frequencies close to, but not exactly at, the peak frequency. This poses a significant challenge because low-cost hardware components are often imprecise. The pass-band width is proportional to the ratio of the Tx wire resistance $R_{coil}$ and inductance $L_{coil}$ values:

$$\Delta f_{width} = \frac{1}{2\pi} \frac{R_{coil}}{L_{coil}}. \tag{6}$$

Because the CSM-EM instrument Tx coil has an inductance of $L_{coil} = 13.28$ mH and a resistance of $R_{coil} = 3.2$ Ohms, the LC filter pass band is $\Delta f_{width} = 38.4$ Hz. While the 38.4 Hz bandwidth combined with a 1.57 kHz peak frequency would,



in theory, prevent the 1.64 kHz RC oscillator signal from generating a sufficient Tx signal strength, we found that these
components did produce high currents (>2.0 A) in the Tx coil. This may be useful when building a new CSM-EM using
substituted parts, because low-cost components with larger uncertainty tolerances still allow the Tx to produce high currents.

### 3.3 Receiver Design

The CSM-EM Rx antenna coil (Figure 5a) has the same dimensions and wire wraps as the Tx antenna, although due to material
heterogeneity the Rx coil resistance and inductance differ slightly from those of the Tx coil. The transmitted signal measured by
the Rx unit is filtered through an LC circuit that can be easily refitted to enable the Rx unit to pass different frequencies. The Rx
coil has an intrinsic inductance $L_{coil} = 12.7$ mH, a resistance $R_{coil} = 2.7\ \Omega$, connected to a tuneable capacitor $C_{coil} = 770$ nF
(Figure 5b). The resulting LC filter has a peak frequency of $f_{coil} = 1.61$ kHz and a pass-band width of $\Delta f_{width} = 33.8$ Hz due
to the finite resistance of the Rx coil.

The filtered Rx signal (modeled in Figure 3a) is amplified using an inverting amplifier configuration powered by two 9 V
batteries (see equation 4). Figure 5b illustrates the amplifier setup with a gain $G = 100$, set by $R = R_1$ and $R_f = R_F$. $R_F$ is a
variable resistor that allows the Rx gain to be changed without hardware refits. The op-amp gain increases the signal amplitude,
allowing the user to observe minute signal changes on a readable scale. The gain value was chosen because it shifts the Rx
input signal up by two orders of magnitude, making any changes in the secondary field more easily observable on a 0.1 mV
resolution multimeter (shown in Figure 3b). The filtered and amplified signals are measured as an RMS voltage by a DMM
connected across the amplifier output.

### 3.4 Construction and Cost Considerations

Table 1 presents the itemized cost for all instrument components. The only tools and resources required for the build are a
soldering iron, wire strippers, electrical tape and zip ties for structural integrity, and containers for electrical modules. The
basic components used in the device build allow the user to substitute most components for those available, which has the
potential to significantly lower the build cost. In most situations, the largest cost contributor - wire - can be re-purposed from
other sources. While the OPA549 power amplifier unit can be built from scratch (Figure 4b), prefabricated boards likely provide
greater Tx system durability, improve the modularity of the Tx design, and facilitate refits and repair.

### 4 Validation

The CSM-EM design goals specify that the system can function at frequencies and Tx-Rx offsets similar to those found in
commercially available FDEM instruments. We used the Geonics EM-34 commercial Tx for validation and comparison. The
CSM-EM system set to function at 1.67 kHz because of its size and compatibility with the Tx frequencies available on the
EM-34. We ensured device functionality through laboratory tests designed to observe whether the CSM-EM could: (1) function
at the desired frequencies; (2) emit stable Tx signals at amplitudes comparable to (or even greater than) the EM-34 Tx; and
(3) detect conductive objects in a free-air environment at reasonable Tx-Rx offsets. After successfully validating functionality



**Figure 5.** (a) Low-cost CSM-EM Rx system. The Rx coil (with attached tuning capacitor $C_{coil}$) acts as a LC bandpass filter for a given frequency. The signal is passed through an inverting amplifier and an RMS voltage is measured by a DMM. (b) Low-cost CSM-EM Rx circuit diagram consisting of the Rx coil/tuning capacitor, an OPA741 inverting amplifier powered by two 9 V batteries, and a DMM.



| Item | Cost (USD) / Unit | # of Units | Cost (USD) / Item |
|------|-------------------|------------|-------------------|
| 14 Gauge Wire | 0.38 / m | 400 m | 152.00 |
| Coil Frame | 12.49 | 2 | 24.98 |
| Breadboard, Wires, Banana Plugs | 12.50 | 1 | 12.50 |
| 9 V Batteries | 2.28 | 2 | 4.56 |
| 12 V Motorcycle Batteries | 29.67 | 2 | 59.34 |
| Switches | 1.20 | 1 | 1.20 |
| Electrical Components | 4.97 | 1 | 4.97 |
| 741 Op-amp module | 1.69 | 2 | 3.38 |
| 549 Power amp Module | 40.00 | 1 | 40.00 |
| Multimeters | 29.97 | 2 | 59.94 |
| | | **TOTAL** | 362.87 |

**Table 1.** Cost breakdown (in USD) for parts required to construct and operate the system. Note that this estimate assumes all products are bought new. Many components (especially wire) can be repurposed, allowing the system to be built at a lower overall cost.

in the laboratory environment, the CSM-EM instrument has been tested in an outdoor setting by performing a survey over an area containing a known conductive anomaly.

## 4.1   Laboratory Validation Tests

The first laboratory trial tested whether the CSM-EM Rx could detect a clear sinusoidal signal at a given frequency with a visible change in the signal amplitude when a conductor was placed nearby. For the trial, the Rx unit was placed in a near 235     free-air environment (i.e., elevated on a mobile stand in the laboratory) at 10 m offset from the CSM-EM Tx. The Tx and Rx were set to function at 1.6 kHz. The brown curve in Figure 6 shows an oscilloscope display of the CSM-EM Rx signal with and without introducing a piece of sheet metal near the Rx. The Rx signal shown is a total field measurement; a superposition of the primary Tx field and secondary fields generated from nearby conductive materials. As shown on the oscilloscope display, the Rx signal had a frequency of 1.67 kHz, with an RMS voltage of 271.9 mV, and a peak-to-peak (Pk-Pk) amplitude of 880.0 mV 240     without the sheet metal present (yellow curve). After the conductive body was placed between the CSM-EM Tx and the CSM-EM Rx, the signal amplitude decreased to less than half of the previous amplitude (brown curve). This is because the secondary field created by conductive bodies destructively interferes with the primary signal, attenuating the total field measured by Rx. The RMS voltage was well within the resolution of a standard DMM. The initial test demonstrated that the Rx measured a stable field very close to the expected Tx frequency with an amplitude of several hundred mV, which is sufficiently observable 245     on commercial DMMs. The test also indicates that the Rx responds to the destructive interference of secondary fields from the conductor, and that the amplitude change can be measured by a DMM as an RMS voltage.



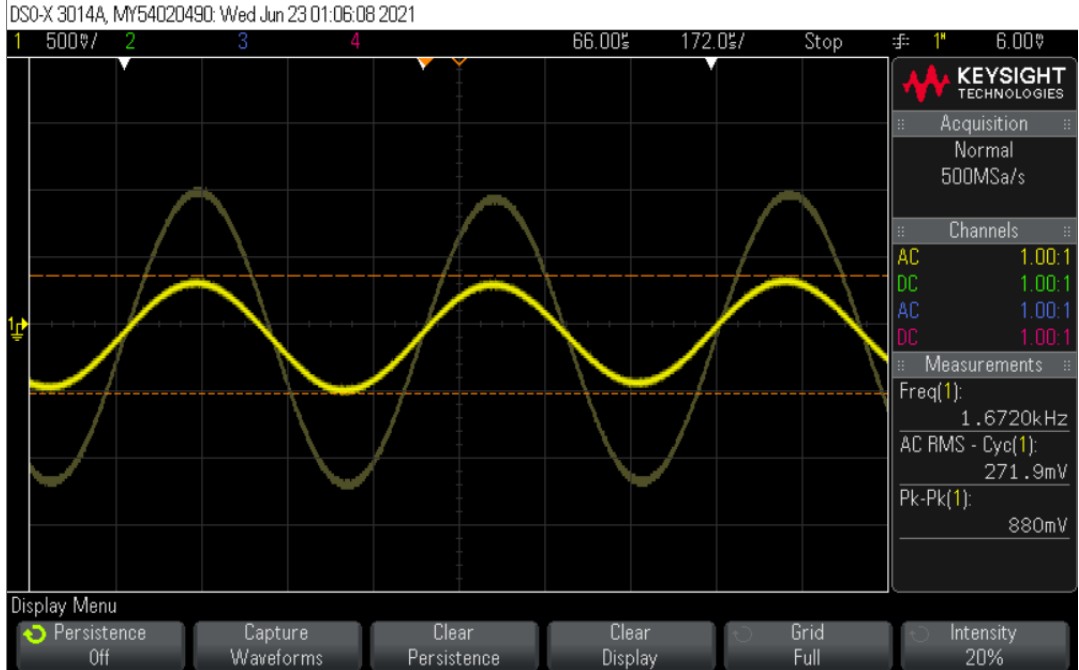

**Figure 6.** CSM-EM Rx signal measured on an oscilloscope using a Rx gain of $G = 100$ at a 10 m Tx-Rx offset with (yellow curve) and without (brown curve) introducing a piece of sheet metal near the Rx antenna. The signal with the sheet metal present is less than half the amplitude of that without metal present. The CSM-EM Tx generated the signal for both measurements.

Next, we tested the CSM-EM Tx unit for its ability to produce a stable signal of comparable strength to that of the EM-34 Tx unit. The system was set up in the same orientation as the former laboratory trial, but using a Rx gain of $G = 50$ at a 5 m Tx-Rx offset. The signal frequency in this trial was, as expected, nearly identical to the measured Rx signal from the previous trial (1.67 kHz). The RMS voltage was 3.42 V and the Pk-Pk amplitude was 9.9 V using the CSM-EM Tx and decreased to less than half the original amplitude when using the EM-34 Tx (yellow and brown curves in Figure 7, respectively), demonstrating that the CSM-EM Tx can produce a stable signal of comparable strength to that of the EM-34 Tx.

### 4.2 Field Validation Tests

Field validation testing involved completing an outdoor field survey over a shallow metallic conductor to test system sensitivity to a conductive object, and a qualitative Tx-Rx offset test. The field survey target was a manhole cover located at the surface in the outdoor Colorado School of Mines Geophysical Discovery Laboratory (GDL). The GDL is a flat sod-covered area with a thin layer of soil 0.5 m deep covering the surface. Backfill underneath the soil continues to a depth of 2 m, with thick shale units extending deeper into the either (Weimer, 1973). While the GDL can be considered geologically homogeneous for near-surface FDEM surveying applications, there is a large amount of electrical and plumbing infrastructure running throughout the area.





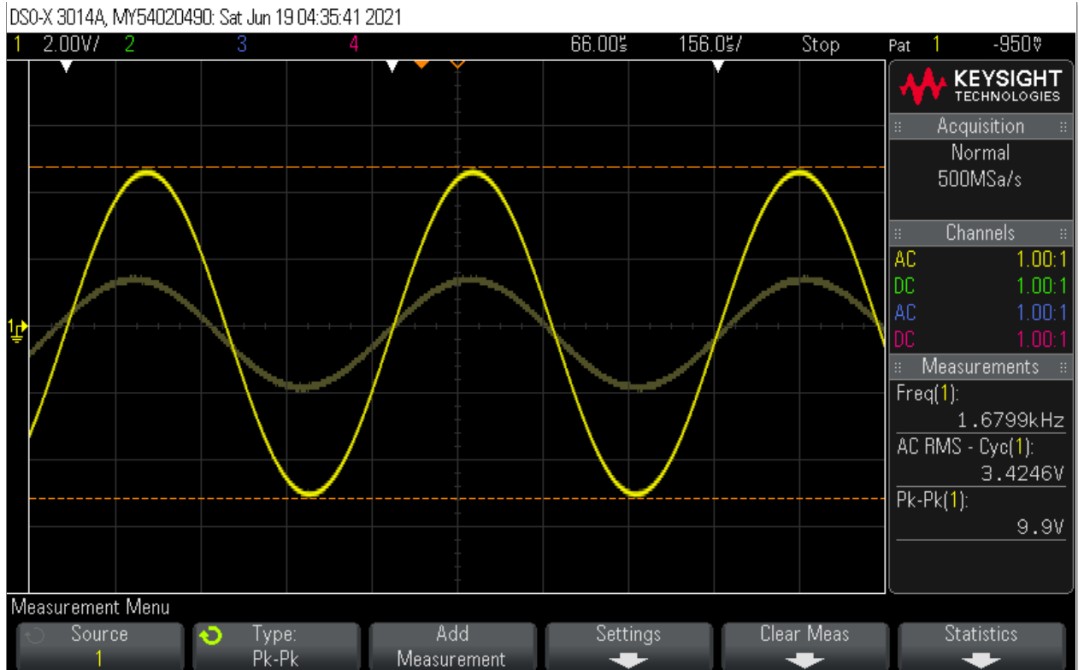

**Figure 7.** Low-cost CSM Rx signal (attached to oscilloscope) using an amplifier gain $G = 50$ at a 5 m Tx-Rx offset with the EM34 Tx (brown) and low-cost CSM-EM Tx (yellow). The amplitude of the signal received from the CSM-EM Tx is over twice that of the EM34 Tx.

The survey consisted of two primary transects: (1) an east-west test line over the manhole cover, and (2) a parallel control line 4 m to the north of the test line (Figure 8a). The first trial used the EM-34 Tx with the CSM-EM Rx on the test line. The second trial combined the CSM-EM Tx and Rx units again on the test line. Finally, a third trial used the combined CSM-EM Tx-Rx system but on the control line for calibration purposes. All Tx and Rx units were set to function at 1.6 kHz. The CSM-EM system for all trials was oriented along the two survey lines shown in Figure 8a with the Tx and Rx units separated

by a fixed $5.0$ m offset and starting at $25.0$ m and $30.0$ m easting, respectively. Measurements proceeded westward every $0.5$ m and stopped when the Tx unit reached the $0.0$ m easting coordinate. The CSM-EM Tx current and Rx RMS voltage were measured at each survey location. The current measurements ensured that the CSM-EM Tx unit exhibited a stable current output throughout the survey. Figure 8b presents RMS voltages measured by the CSM-EM Rx and shows a significant anomaly detected in both the EM-34 Tx and CSM-EM Tx trials (blue and orange curves, respectively) when either the Tx or the Rx

passed directly over the target. In comparison, the CSM-EM control line data (green curve) exhibited no significant voltage drop. Figures 8c shows the current in the CSM-EM Tx over the manhole cover line (blue curve) and over the control line (green curve).

The Tx-Rx offset testing was conducted in the GDL area away from the manhole cover, so that measurements could be made over a homogeneous subsurface. The CSM-EM Rx output was observed using an oscilloscope to gauge signal stability and a

DMM to measure RMS voltage. Both the Tx and Rx devices were set to the same system parameters from the manhole cover







**Figure 8.** Geometry and results from the CSM-GDL field validation test using a 1.6 kHz Tx frequency. (a) Survey geometry showing the 0.75 m diameter manhole cover surface target located at 18.5 m easting on the test line. The control line is parallel to and 4 m North of the test line. The only other known nearby conductive body is a sprinkler box located close to the survey area. (b) The orange and blue curves show the test line data for the EM-34 Tx and CSM-EM Rx and the CSM-EM Tx and Rx combinations, respectively. Both curves show significant RMS voltage reductions when either the Tx or Rx passes over the target. The green curve presents the control line data unaffected by the surface metal objects. (c) The blue and green curves show CSM-EM Tx current for the test and control lines, respectively. Along the test line, the Tx current dips when the antenna is directly over the manhole cover, as well as at 6.5-7.0 m easting.

field test. The CSM-EM Tx and the EM-34 Tx were tested separately, with the CSM-EM Rx in an HCP coupled orientation. Measurements took place at 5.0 m intervals. The EM-34 Tx paired with the CSM-EM Rx was tested starting at a Tx-Rx offset of 10.0 m, and continuing until the sinusoidal Rx signal was no longer resolvable on the oscilloscope. At 10.0 m the EM-34 Tx generated a clean signal with an RMS voltage of 150 mV, representing total field strength. When the system was moved to a

15.0 m offset, the Rx signal was very weak and had an RMS voltage of less than 60 mV. Because the CSM-EM Tx generated a larger signal in laboratory testing compared to the EM-34 Tx, the CSM-EM Tx coupled with the CSM-EM Rx was tested at





larger offsets. At 20.0 m, the CSM-EM Tx generated a clean signal with a 130 mV RMS voltage, comparable to the EM-34 Tx signal at 10.0 m offset. At 25.0 m the CSM-EM Tx signal was still resolvable from the background noise, but the RMS voltage had dropped to 80 mV. At 30.0 m offset, the CSM-EM Tx signal was very weak, with an RMS voltage of 60 mV. The offset

test demonstrated that the CSM-EM Tx can produce a resolvable Rx signal at up to 25.0 m offset from the CSM-EM Rx unit.

## 5    Discussion

The CSM-EM system provided a variety of results from the engineering design process and the prototype validation tests. We initially designed the CSM-EM unit to be a tilt-angle EM system prototype (Nabighian and Corbett, 1991); however, early laboratory tests suggested that the Tx-Rx system was capable of conducting amplitude-based FDEM measurements when

connected to a DMM measuring RMS voltage. While the laboratory and field test results present a proof of concept that the CSM-EM unit is capable of acquiring field-scale FDEM data, they also highlight several challenges experienced during CSM-EM data acquisition and analysis as well as opportunities for further research.

### 5.1    Proof-of-Concept

The CSM-EM unit can function at multiple frequencies at field-scale Tx-Rx offsets in an outdoor environment with a Tx signal

strength comparable to that of the EM-34 Tx. The low-cost components required to construct the system are easily procurable; many parts also could be substituted to further decrease costs. The CSM-EM system is easy to adapt to a wide range of Tx-Rx frequencies with minor hardware changes, and can acquire data using a voltmeter. The system measurements can be connected to the analog-to-digital converter available on most microcontrollers (e.g., an Arduino UNO) for cost-effective, light-weight digital recording.

Field testing demonstrated that the complete CSM-EM system is sensitive to conductive objects and can produce similar amplitude-based data as the EM-34 Tx and CSM-EM Rx combination. The CSM-EM field trials show that the CSM-EM Tx produces larger Rx voltages and more prevalent anomalies than the commercial Tx. The CSM-EM Tx-Rx data have a variance of $0.118\ V^2$ over the test line, excluding data points over the target, and a variance of $0.029\ V^2$ over the control line. The data for the EM-34 Tx with the CSM-EM Rx have a variance of $0.011\ V^2$ over the test line, excluding data points over the target.

The higher variance of the data measured using the CSM-EM Tx is significant enough to prevent the system from detecting small-scale near-surface resistivity changes, such as those from geology or fluid content. The CSM-EM Tx current (Figure 8c) exhibits similar trends to the CSM-EM Rx voltage when the Tx is directly over the manhole cover. The Tx current signal showed a variance of $0.012\ A^2$ over the control line and $0.036\ A^2$ over the test line. Along with higher data variance using the combined CSM-EM Tx/Rx, limited sampling of DMM readings make any kind of noise analyses difficult.

Noise present in the CSM-EM field data indicates a lack of stability in the Rx signal and Tx current, which may be due to numerous factors including the lower build quality and reduced rigidity of the CSM-EM antennas, as well as the precision of the low-cost hardware (e.g., resistors, capacitors, inductors, repurposed wiring) used in the CSM-EM Tx build. The CSM-EM antennas were constructed around plastic liners and easily deform, which can affect the stability of signal transmission and





could account for the greater observed variance in the Tx and Rx antennas. (The antennas were held flat in the HCP orientation
to mitigate these effects during trials.) The quality and tolerances of the hardware used in the CSM-EM Tx unit may have
affected signal quality during testing. Issues with the hardware component connections could be mitigated by integrating the
components into a printed circuit board.

## 5.2 Future Development

The testing performed on the prototype CSM-EM system allows for several future developments. While the CSM-EM system
resembles the EM-34 in size and shape, next-generation designs could use the same circuitry on a miniaturized scale to create
a system more similar in size to lightweight single-operator EM systems. A miniaturized system would require less material
(specifically antenna wire) and could function as a single rigid structure. Along with miniaturization, the Tx and Rx tuning
capacitors could be attached to rotary switches, which would permit the frequency range of the device to be changed without
any hardware modifications.

Microcontroller inboard analog-to-digital converters are capable of voltage and current measurements when combined with
basic electrical components (ana). This would allow for the CSM-EM system to digitally record measurements. Including a
microcontroller module (e.g., an Arduino or Raspberry Pi) opens up the possibility of future CSM-EM versions to be operated
remotely which, when combined with the system's potentially lightweight nature, opens up opportunities for multi-Tx-Rx
experiments.

By combining a more robust build with rapid autologger sampling, we aim to decrease noise during data acquisition and
better analyze more densely recorded data through processing. With an improved system, we can determine whether the CSM-
EM design is sensitive to small-scale conductivity variations due to geology or subsurface fluid content. Modifications are
underway to perform phase-based measurements for a more direct comparison to commercial instruments. These mechanical
and data-collection improvements will dictate future applications of the CSM-EM system in near-surface geophysical investi-
gations.

## 6 Conclusions

The primary goals of this study were to design, build and validate a low-cost transmitter-receiver FDEM system for un-
der US\$500 that is of comparable size and signal transmission strength to commercial grade systems. The CSM-EM device
costs US\$363 for the current design when using all new parts and can detect conductive objects in a field environment using
amplitude-based signal measurements. The modular design of the unit allows users to easily replace components and to replace
the DMM system with a microcontroller-based autologger. The CSM-EM functions at a variety of frequencies with minimal
hardware adjustments and produces a stable signal of comparable strength to commercial systems (e.g., the Geonics EM-34
Tx). This proof-of-concept device provides a foundation for the future development and use of low-cost FDEM systems for
near-surface geophysical and other related investigations.



*Author contributions.*  GW, JC, JA, and AS designed, built, and tested the instrument. GW and JS were responsible for the preparing and writing and preparing the manuscript with contributions from other co-authors. AS and JS were responsible for completing manuscript revisions.

*Competing interests.*  The authors declare that they have no conflict of interest.

*Acknowledgements.*  To be completed at a later time.





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
