# Peer review of "Developing a low-cost frequency-domain electromagnetic induction instrument"

_EGUsphere, 2022_

## Referee Comment (RC2)

390

[referee-annotated manuscript omitted]

---

## Author Response (AR1)

Hello, thank you for taking the time to read and review the manuscript. Your comments have been valuable in the development and improvement of our paper. Our specific responses addressing each comment are included below:

**Figure 1:**

- Comment: I might be confusing things here, but based on the direction of the arrows for the eddy currents (orange lines), shouldn't the secondary field (dashed black lines) have opposite arrow directions than the ones shown in the figure?
- Response: The arrow directions of the secondary field were indeed flipped. This has been fixed in the edited document.

**Line 93:**

- Comment: I often wonder about this assumption. Is it really valid to say that magnetic permeability is equal to the free space value? Wouldn't the conductivity anomalies of interest have a relative magnetic permeability? I know for many materials the relative magnetic permeability is close to 1 but there are many minerals (and background subsurface situations) for which this may not be true. Any way to account for this?
- Response: There are definitely some materials for which this assumption would not be valid. The free-space value was chosen in the modeling to gauge the basic design expectations we had for the system, specifically for small-scale research and future humanitarian applications. We hope to expand our modeling parameters and our testing to other subsurface environments to observe the CSM-EM response.

**Lines 104-106:**

- Comment: I don't quite understand the distinction pointed out here. Changing any of these parameters will change the voltage measured across the receiver.
- Response: Our goal was to distinguish between parameters that are set during the instrument design phase (antenna dimensions, etc.) and parameters that can be set/changed during a survey (offset, functioning frequency of the system).

**Equation 1:**

- Comment: Is this valid for far-field, near-field, or both? Any comment on the relative size of offset (r) with respect to the TX loop radius (i.e. Is this valid when the TX looks like a point dipole to the RX)?
- Response: We are using the general equation, valid for both near and far-field situations, for calculating the vertical magnetic field.

**Line 113:**

- Comment: "… which measures the strength of the total vertical magnetic field…"
  Are you measuring the strength of the field, or the rate of change of the field?
- Response: We are measuring the time rate of change of the field, and have
  clarified the statement in line 113 to "...which measures the time rate of change of
  the total vertical magnetic field… "

**Line 117:**

- Comment: To be consistent throughout the paper, you should use either
  resistivity or conductivity.
- Response: For the edited manuscript we will be using resistivity.

**Line 118:**

- Comment: "…to detect for most half-space resistivities" in the presence of the
  primary field.
- Response: An important distinction to make, this has been added to the
  document.

**Figure 3:**

- Comment: I am a bit confused here. From what I understand, you are subtracting
  the voltage due to the primary field (a constant at each offset) from the total
  voltages of figure (a) to get the results of figure (b). Correct? If so, why does the
  order of the curves change between figures (a) and (b)? The primary field is a
  constant for all the different cases of the halfspace resistivity, so subtracting it
  from the curves of figure (a) shouldn't change the relative order of the curves.
  What am I missing here?
- Response: The only large-scale change in the order of the curves is present
  between the perfect conductor and the other halfspace curves. We have added a
  'perfect resistor' curve, which is identical to the primary field. Since the difference
  between the calculated primary field and the total field curves for the half-spaces
  is very small, the secondary field decay curves are significantly smaller than their
  total field counterparts and have a reversed ordering.
- Comment: The y-axis is voltage plotted in log scale. Therefore, the label should
  be voltage and NOT Log10 voltage. Plots (a) and (b) should be labeled "Total
  field" and "Secondary field" respectively for clarity. The y-axis label for both plots
  can then be changed to Voltage. Right now, the label "Log10 Voltage difference"
  is incorrect and confusing.

- **Response:** Axis labels and title edits have been made, please see attached Figure 3a/b for changes
- **Comment:** The x-axis which is offset should have some reference to the radius of the TX and RX dipoles. Are your TX and RX dipole radii small enough that all the values on the x-axis are far-field? If not, then your offset should be in units of (r/R) where "r" is offset and "R" is the TX dipole radius.
- **Response:** The 0.3 m TX and RX radii are significantly smaller than the minimum 5 m offsets we used in field testing and are also smaller than the offsets used in the modeling shown in Figure 3.
- **Comment:** In plot (b) what is happening with the case of the 1 Ohm.m halfpace? What is the strange dip at offset between 40 and 50 m? Why does this curve cross the 10 Ohm.m and 100 Ohm.m curves? This is a prominent feature and is not explained.
- **Response:** There is a sign reversal taking place at ~42 m along the 1 Ohm.m halfspace. This is the distance at which the total field and the primary field responses are closest in value. During modeling, we found that the higher resistivity halfspace response curves showed similar trends at larger offsets.
- **Comment:** In your plot (a) it might be good to also show the case of the perfect resistor as well (i.e. free-space or the primary field that you subtract in figure (b)).
- **Response:** A perfect resistor curve has been added to 3a to represent the primary field we are subtracting to create 3b.

**Line 143:**

- **Comment:** Why did you measure the inductance in this way? Did you not have an inductance meter, or was there an advantage to doing it this way?
- **Response:** We did not have easy access to an inductance meter and wanted to ensure a precise measurement for the Tx/Rx coil values.

**Line 200:**

- **Comment:** There needs to be an explanation of why this is the case? Is it because the resistance and inductance are not exactly what you measured? I suggest including a plot of gain (dB) vs. frequency around the central frequency of 1.57 kHz for your TX transmitter stage.
- **Response:** A figure has been added (see attached) illustrating the peak frequency and filter width of the RLC circuits in the Tx and Rx antennas.

**Figure 5(b):**

- **Comment:** What is the point of switches S1 and S2? Is there a need to switch power on and off selectively to either side of the op amp?

- Response: The switches were added so that we could disconnect the batteries from the receiver without physically detaching them. There was no need to selectively turn off either side of the op-amp, and the switches could be replaced with a double pole single throw (DPST) switch if available. The SPST switches were used because we have found that, in many situations, an SPST is much easier to procure in most regions.

**Lines 240-241**:

- Comment: Please double check that you have the yellow and brown curves correctly identified here. From your description it seems that they should be flipped.
- Response: Thank you for that catch - the colors in the description were switched. This has been fixed.

**Line 270:**

- Comment: This description does not match what is shown in figures (b) and (c).
- Response: The labeling for figures 8b and 8c were switched by mistake, this has been fixed.

**Figure 8:**

- Comment: The figure caption description does not match figures (b) and (c). Why is the y-axis of figure (b) labeled current (A)? This does not seem to match the figure caption.
- Response: For both comments - the captions for figures 8b and 8c were switched by mistake. This has been remedied in the edited version of the manuscript

**Lines 273-285:**

- Comment: This paragraph is describing something that should be shown in a figure. Please add a figure for the offset testing.
- Response: We hope to do more offset testing with the development of the second prototype CSM-EM (under construction), and would like to build an offset graph with a more dense dataset in the future.

**Final comments:**

- Comment: Lastly, does the operator of this instrument have to subtract the theoretical value of the primary field from the RX measurements in order to work

with the data, or do you envision the operator using the total field data directly as it is?

- Response: For our current purposes, we envision the operator using the total-field amplitude data as is (similar to that shown in our field trial). Once we obtain more dense time-series sampling on the device (using a microcomputer autologger or otherwise), we hope to work directly with the secondary field (and eventually consider both amplitude & phase, or in-phase & quadrature).